# Influence of LINE-Assisted Provision of Information about Human Papillomavirus and Cervical Cancer Prevention on HPV Vaccine Intention: A Randomized Controlled Trial

**DOI:** 10.3390/vaccines10122005

**Published:** 2022-11-24

**Authors:** Yu Ota, Kyoko Nomura, Nozomi Fujita, Tomoya Suzuki, Makoto Kamatsuka, Natsuya Sakata, Kengo Nagashima, Junko Hirayama, Naoko Fujita, Kuniko Shiga, Noriaki Oyama, Yukihiro Terada

**Affiliations:** 1School of Medicine, Faculty of Medicine, Akita University, Akita 0108543, Japan; 2Department of Environmental Health Science and Public Health, Akita University Graduate School of Medicine, Akita 0108543, Japan; 3Biostatistics Unit, Clinical and Translational Research Center, Keio University Hospital, 35 Shinanomachi, Tokyo 1608582, Japan; 4Laboratory of Plant Physiology, Department of Biological Production, Faculty of Bioresource Science, Akita Prefectural University, Akita 0100195, Japan; 5Japanese Red Cross Akita College of Nursing, Akita 0101493, Japan; 6Department of Gynecology, Akita Red Cross Hospital, Akita 0101495, Japan; 7Department of Obstetrics and Gynecology, Akita University Graduate School of Medicine, Akita 0108543, Japan

**Keywords:** human papillomavirus vaccine, HPV, Japan, vaccine acceptability, college/university students, vaccine intention, knowledge, health literacy, health belief model

## Abstract

We conducted a prospective, randomized two-arm, parallel group, and open label trial to investigate whether the use of LINE would increase HPV vaccine intention among not completely vaccinated university students. In June 2020, we recruited students aged between 18 and 35 years from four universities in Japan. Among the 357 enrollees (female, 53%), 178 and 179 participants were randomized into the LINE and Mail groups, respectively. At baseline, within three years, vaccine intention was observed in 40% vs. 42% of participants, respectively. At the first intervention, which provided similar PDF leaflets about HPV vaccine and cervical cancer prevention, there was no significant difference in vaccine intention between the two groups. However, at the second intervention of LINE-assisted knowledge intervention for 5 days per week for 7 weeks, the LINE group had a higher proportion of vaccine intention than the no intervention group (66% vs. 44%, OR: 2.62, 95% confidence interval (CI): 1.59–4.35) in per-protocol analysis. The significance remained in the intention-to-treat analysis of multiply imputed datasets. Although LINE did not directly increase HPV vaccine intention compared to conventional posts, the LINE-assisted provision of information was effective in improving HPV vaccine intention among Japanese university and college students.

## 1. Introduction

Under the Japanese Immunization Program in 2013, the human papillomavirus (HPV) vaccine program was launched for girls aged between 12 and 16 years [1]. At the time, two types of HPV vaccine with three doses of immunization were available, including the bivalent HPV vaccine (Cervarix, GlaxoSmithKline, Rixensart, Belgium) with a 0–1–6 month scheme and the quadrivalent HPV vaccine (Gardasil, MDS, Lyon, France) under a 0–2–6 month scheme [2]. Gardasil was approved for male vaccination in December 2020 [3]. However, as of now, men are not eligible for routine HPV vaccination in Japan. Although the program for girls had been enacted, only a few months after the first reports of adverse reactions surfaced, the Ministry of Health, Labour, and Welfare decided to suspend active recommendation in 2013 [4]. The vaccination coverage was as high as 70% during the period of active recommendation [5], but later dropped to less than 1% [6]. Since then, there has been much controversy over the resumption of vaccination activities [7,8], and the whole world watched Japan’s stagnation. However, in April 2022, the Ministry suddenly decided to resume active recommendations. Relevant sources have underscored the importance of catching-up for the girls who missed the active recommendation between 2013 and 2021 [7]. However, it is questionable whether those who were exempted from the recommendation would choose to receive the HPV vaccine at their own expense.

It has been previously reported that knowledge, health literacy, health belief models, and awareness were important promotors of HPV vaccine intention [9,10,11,12]. A recent systematic review of studies examining the correlates of HPV vaccine uptake among teenage girls reported that higher vaccine-related knowledge was positively associated with HPV vaccination [13]. Furthermore, health literacy does not seem to be directly linked with vaccination [14] but may be linked with HPV knowledge [15,16]. Studies based on the health belief model have identified the lack of information and knowledge about the HPV vaccine, low risk perception of HPV infection, side effects, and cost of HPV vaccination as barriers to HPV vaccination [17,18,19,20].

Today, social media is central to communication among young people, and hence it is highly expected that the use of social media may influence HPV vaccine uptake [21,22]. Social media is featured from the cost-effectiveness point of view [23]. A systematic review showed that social media affects HPV vaccination uptake [24], and another review showed that social media improved HPV knowledge but did not improve HPV vaccination uptake [25]. According to a scoping review, such inconsistency may be explained by the ambiguous definition of social media [26]. Indeed, there are social media applications, such as Twitter, Facebook, and YouTube, that promote HPV vaccine uptake [24,25]. We hypothesized that social media and frequent interventions that provide information and opportunity to communicate with companies, friends, or health professionals are key factors to increase HPV uptake. Hence, this study has two objectives: (1) we investigate whether providing information by LINE, the most popular communication tool for young people in Japan, compared to Mail, a letter or document sent by post, increases HPV vaccine intention, HPV knowledge, HPV health literacy and cervical cancer prevention, and health belief models; and (2) we investigate whether LINE-assisted frequent interventions that provide information and opportunity to communicate with companies improves HPV vaccine intention, knowledge, health literacy, and health belief models compared to the no intervention group among Japanese young adolescents who had missed the active recommendation of the HPV vaccine by the government.

## 2. Materials and Methods

### 2.1. Study Design

This study was a prospective, randomized two-arm, parallel group, and open-label trial with equal allocation to each arm (see Figure 1).

### 2.2. Study Participants

We recruited students from three universities (one national university with a medical faculty and two public universities) and one nursing college in Akita Prefecture, the northern part of Japan. We recruited participants during the first round (Round 1) between June and December 2020, and the second round (Round 2) between April and June 2021. The inclusion criteria were as follows: a male or female student aged between 18 and 35 years, a student who belonged to any of the four schools, and who had not been vaccinated. In the Akita prefecture, the majority of university and college students were aged 18–22 years and had missed active recommendations because they were ineligible for the publicly subsidized HPV immunization program in 2013 due to their age.

Participants who were not of Japanese ethnicity and those who did not use LINE were excluded. In Japan, at the time of this study’s investigation, LINE was the most popular social networking service for young people to communicate with friends, family, school mates, and any other community members. The recruitment methods included campus electronic bulletin boards that addressed individual students, classroom websites, and on-campus notice boards. The students accessed the Google form by scanning the QR code attached to the flyer and read the outline of the study protocol. The purpose of the study and survey procedures were also included on the website of the Department of Public Health at Akita University. If a student agreed to participate, they submitted a consent form to the research administrative office; these forms were initially sent by e-mail from the office. Immediately after the submission of the informed consent form, they were asked to answer a questionnaire created in Google forms using a link attached to an e-mail from the office. The questionnaire was sent at baseline, and during the first and second interventions. We sent reminder e-mails and called at least twice on each occasion. The participants were given a JPY 500 (USD 5) equivalent gift card for each questionnaire. If the participants completed all the questionnaires, they received three gift cards of JPY 500 (USD 5).

Participants were randomly assigned to the LINE and Mail groups in a 1:1 ratio (experimental or control group) by an independent administrator (centralized randomization). The randomization with a sex-stratified block size of 4 and 6 was generated beforehand by a statistician (not involved in the enrolment and intervention). Researchers, except for the statisticians (Kengo Nagashima and N.F.), were blinded to the randomization and allocation. We created a private open chat in which the LINE participants were regulated by the administrative office under an anonymous nickname and asked the participants to follow the ground rules for concealed allocation, which were: (1) not to talk outside the chat room and (2) not to talk to anyone about their participation in this study. Participants were asked to not talk to other LINE participants until the second intervention, interaction between enrollees was permitted at the second intervention, and the detailed study aims related to the allocation were not disclosed to the participants. Randomization and intervention were conducted twice according to Rounds 1 and 2.

The study was approved by the Research Ethics Committee of Akita University (No. 2387), and it follows the ethical international standards established in the Declaration of Helsinki. Written informed consent was obtained from each participant prior to the study, and the present trial was registered at the University hospital Medical Information at Tokyo University with the registration number: UMIN000044750.

### 2.3. Intervention

We conducted the first intervention to observe whether there was a difference in HPV vaccine intention, HPV knowledge, health literacy, and health beliefs between the LINE and Mail groups. First, we sent an A1 sized two-page leaflet on HPV and cervical cancer prevention to both groups. The leaflet was created by the authors by referring to the relevant sources from the Ministry of Health, Labour, and Welfare [27]; Japan Pediatric Society [28]; Japan Society of Obstetrics and Gynecology [29]; two websites operated by the Association for the Dissemination of Information on HPV [30]; and by the media for nursing students and young female nurses [31] in Japan. The leaflet contained brief information on what HPV is and on cervical cancer prevention. In the LINE group, we used LINE as an information carrier and posted the leaflet in PDF format. In the Mail group, we sent its printed version. We compared HPV vaccine intention, HPV knowledge, health literacy, and health beliefs at the first intervention.

Subsequently, we conducted the second intervention to observe whether there was a difference in HPV vaccine intention, HPV knowledge, health literacy, and health beliefs between the intervention and no intervention groups. During this period, we sent information on HPV and cervical cancer prevention 5 days a week for a period of 7 weeks only to the LINE group (i.e., the LINE-assisted intervention group), and considered the Mail group as the no intervention group. Participants in the LINE-assisted intervention group were instructed in advance that they would be able to consult or tweet freely within the LINE community specifically created for this purpose. The administration office issued ground rules before each participant entered the chat room, which prohibited any critical, intimidating, or harsh comments. Membership would be forcefully terminated if there was no change in behavior after a warning had been issued for any offence. All the participants were kept anonymous and allowed to communicate with each other freely at any time during the second intervention. The administration office for this specific LINE closed group consisted of five medical students (Y.O., T.S., N.F., M.K. and N.S.) and one faculty (K.N.), who were also members of this chat room. The five members dealt with any queries or comments and escalated more complex issues to the faculty. The medical students received basic training on potential interactions or concerns with participants and held a meeting at least once a week. The image of the intervention is shown in Figure 2. All the dialogues were recorded by assigning specific numbers and were grouped according to content. The authors also created the content for the 7-week LINE intervention by referring to relevant sources. This included the public health issue of cervical cancer (i.e., statistics of mortality and anatomy of uterine and disease characteristics) during the 1st week; what is HPV and how it is related to uterine cervical cancer during the 2nd week; prevention strategies for cervical cancer and how the HPV vaccine targets cervical cancer during the 3rd week; safety and efficacy of the HPV vaccine in the 4th week; current status of HPV vaccination in Japan during the 5th week; types, doses, price, and national financial compensation system for adverse effects during the 6th week; and places to get shots and other relevant links during the 7th week.

The participants completed the Google form-based surveys at three time-points: the baseline, first intervention, and second intervention.

### 2.4. Measures

We asked the participants about HPV vaccine intention, HPV knowledge, health literacy, and health beliefs related to the HPV vaccine and cervical cancer prevention using a Google form-based questionnaire at the baseline and at the first and second intervention. The list of questions in the Google form-based questionnaire can be seen in Appendix A. A month after we provided the information, the Google form-based questionnaire was distributed (the first intervention). Furthermore, one month after the 7-week 5 days provision of information, the Google form-based questionnaire was again distributed (the second intervention, Figure 1).

#### 2.4.1. HPV Vaccine Intention

This study’s primary outcome is the students’ intention to receive the HPV vaccine. Thus, the study participants were asked the question, “At what time do you intend to receive the HPV vaccine?” They answered based on a five-point scale: “immediately” = 1, “within six months” = 2, “within a year” = 3, “within three years” = 4, “no intention to receive a vaccine”, or “not sure” = 5. As this study was conducted during the COVID-19 pandemic, there was only a small percentage of participants who answered “immediately.” Subsequently, because of statistical reasons, the response patterns were grouped into binary with “immediately” to “within 3 years” as one group and the rest as “otherwise.” This study’s secondary outcome is HPV knowledge, health literacy, and health beliefs related to the HPV vaccine and cervical cancer prevention.

#### 2.4.2. HPV Knowledge

We asked the participants to answer 20 knowledge questions on the HPV vaccine. The first 10 questions were basic, but the next 10 questions would require that students study individually by collecting further accurate information. The question items from 1–10 were as follows: (1) HPV is a human immunodeficiency virus (correct answer: incorrect); (2) There are four types of HPV (incorrect); (3) HPV causes cancer of the uterine body (incorrect); (4) HPV cannot be transmitted by a single sexual intercourse (incorrect); (5) HPV can only be transmitted by males, not by females (incorrect); (6) Types 6 and 11 cause more than two-thirds of all cervical cancer (correct); (7) Cervical cancer can be prevented by using vaccines (correct); (8) Cervical cancer screening is done by cytology (correct); (9) The eligible age for the HPV vaccine to be taken at public expense is up to 35 years old in Japan (incorrect); (10) Cervical cancer is caused by persistent infection with high-risk HPV (correct). The response patterns for the first 10 questions were either “correct,” “incorrect,” or “do not know.” The answer to question (8) has now changed in Japan, but other countries use HPV typing or cytology and HPV typing. The question items from 11–20 were as follows: (11) What do you think is the screening rate for cervical cancer in Japan? (1. 80% or more, 2. 60–80%, 3. 40–60%, 4. 20–40%, 5. 20% or less) (correct answer: 4); (12) In some foreign countries, HPV vaccination is also given to males (1. correct, 2. do not know, 3. incorrect) (correct answer: 1); (13) What percentage of females with a history of sexually transmitted infections do you think will experience HPV infection by the time they are 50 years old? (1. 80% or more, 2. 60–80%, 3. 40–60%, 4. 20–40%, 5. 20% or less) (correct answer: 2); (14) Which types of HPV are less likely to cause cancer? (1. type 16, 2. type 18, 3. type 52, 4. type 31, 5. type 2) (correct answer: 5); (15) What is the leading cause of cancer death among females in their 20 s and 30 s? (1. lung, 2. stomach, 3. liver, 4. uterus, 5. breast) (correct answer: 5); (16) It is necessary to have a regular checkup for cervical cancer even if you have HPV vaccine (1. correct, 2. do not know, 3. incorrect) (correct answer: 1); (17) How effective do you think the cervical cancer vaccination is in preventing cervical cancer? (1. 100%, 2. 80–99%, 3. 60–70%, 4. 50% or less) (correct answer: 2); (18) How often do you think serious adverse reactions will occur due to vaccination compared to flu vaccine? (1. low, 2. about the same, 3. somewhat high, 4. extremely high) (correct answer: 2 or 3); (19) What do you think is the probability for the occurrence of cervical cancer in the lifetime of a female? (1. one in two persons, 2. one in 12 persons, 3. one in 75 persons, 4. one in 155 persons) (correct answer: 3); (20) Which of the following do you think is used to confirm the diagnosis of cervical cancer? (1. cytology, 2. hemolysis, 3. HPV virus type, 4. colposcopy (histology), 5. Pap smear) (correct answer: 4). The total score was divided into binary at median.

#### 2.4.3. Health Literacy

Considering a previous study [32], we investigated health literacy regarding the HPV vaccine based on the following five questions: (1) Are you able to collect information related to the HPV vaccine and cervical cancer screening from various sources, such as newspapers, books, TV, and the Internet? (2) Are you able to select the information that one wants from a large amount of information related to the HPV vaccine and cervical cancer screening? (3) Are you able to consider the credibility of the information related to the HPV vaccine and cervical cancer screening? (4) Are you able to understand and communicate information related to the HPV vaccine and cervical cancer screening to others? (5) Are you able to decide on plans and actions to improve health based on the information about the HPV vaccine and cervical cancer screening? The questions were asked using a six-point Likert scale, asking the respondents to choose an option from the following: (1) very easy, (2) somewhat easy, (3) Intermediate, (4) somewhat difficult, (5) very difficult, (6) do not know/does not apply. Each item was recalculated by subtracting from “6” ranging from 1, “strongly difficult” to 5, “strongly easy” without counting on “do not know/does not apply”, indicating that a higher score is more likely to have a higher health literacy. The total score was divided into two groups at the median score.

#### 2.4.4. Health Beliefs

We exploited the health belief model [33] to identify the factors related to vaccine intention and created the following statements in relation to HPV vaccine and cervical cancer: (1) I regard myself as susceptible to HPV infection (perceived susceptibility), (2) I believe it would have potentially serious consequences if I get cervical cancer (perceived severity), (3) Cervical cancer is a life-threatening disease (perceived severity), (4) Vaccination would reduce the susceptibility or severity or lead to other positive outcomes (perceived benefit), (5) The price of the vaccine is too expensive (perceived barriers), (6) I am worried about the side effects of the vaccine (perceived barriers), (7) I am too busy to go for the vaccination (perceived barriers), (8) I do not know where I can get the vaccine (perceived barriers), (9) Three doses of vaccination is too much trouble (perceived barriers), (10) My parents do not agree with the vaccination (perceived barriers). The respondents were asked to choose only one option from the following: (1) agree, (2) somewhat agree, (3) undecided, (4) not really agree, (5) disagree, (6) do not know. The response patterns were further grouped into binary with “agree”, and “somewhat agree” as one group and the rest as “otherwise”.

#### 2.4.5. Covariates

Other covariates measured by the self-administered questionnaire were sex, age, university, faculty (healthcare, including medicine, nursing, science technology, or other), smoking (current, ever, never) and drinking (everyday, a few times a week, seldom, never), sleep duration, exercise volume and intensity based on a unit of metabolic equivalents, frequency of breakfast per week, and the reasons for not getting the vaccine. The reasons for not getting the vaccine included concerns about side-effects, parents’ opposition, I do not know where to get vaccinated, expensive, sexually inactive, I don’t want to bother getting vaccinated (i.e., it is troublesome), I would not get infected, friends not vaccinated, vaccination is unnecessary if I go for regular check-ups (i.e., checkups alone are enough), schools do not recommend, afraid of get infected with COVID-19, and others, and the participants were allowed to choose as many as they had.

### 2.5. Statistical Analyses

Sample size calculation revealed a total sample size of 194, with 10% dropout, α = 0.05, 1-β = 0.8, and 5% difference in proportion. The descriptive analysis included the number and content of the tweet in the LINE open chat during the 3-month follow-up period.

HPV vaccine intention, HPV knowledge, health literacy, and health belief model related to HPV were assessed using a logistic regression model that was stratified by sex at the first (LINE vs. Mail groups) and second (LINE-assisted intervention vs. no intervention groups) interventions. We also assessed the 20 questions on HPV knowledge and 10 questions on health beliefs using a logistic regression model that was stratified by sex at the first intervention. We applied both per-protocol (PP) and intention-to-treat (ITT) analyses using a multiple imputation technique [34] to create and analyze the multiply imputed datasets. The discordant rate of HPV vaccine intention at the baseline, first intervention, and second intervention was compared using the McNemar test. Time series-analyses using the PROC MIXED procedure in SAS were performed on the continuous variable of HPV knowledge, health literacy, and health beliefs within each allocation group and was compared between the baseline, first intervention, and second intervention. Two-tailed tests were used to determine the significance at the 5% level.

All statistical analyses were performed using SAS version 9.4 (SAS Institute Inc., Cary, NC, USA).

## 3. Results

### 3.1. Baseline Characteristics

Among the 395 undergraduate or graduate students who agreed to participate in this study, 38 participants did not meet the inclusion criteria. Furthermore, after excluding 4 students who did not report their age, 357 students were randomly assigned to the LINE (*n* = 178) and Mail (*n* = 179) groups. The participants’ characteristics are presented in Table 1. Among the total 357 eligible university students, there were 189 females (53%) and 168 males (47%) with a mean age of 20 years. The 25th percentiles, median, and 75th percentiles of age were 19 years, 20 years, and 22 years, respectively. There were 349 participants under the age of 26 (98%) and 8 over the age of 27. Less than half (41%) of the total number of participants belonged to the health care faculty. The most frequent reasons for not getting the vaccine were both “concerns about side-effects” (*n* = 86, 24%) and “parents’ opposition” (*n* = 86, 24%), followed by “I do not know where to get vaccinated” (*n* = 79, 22%), and “expensive” (*n* = 53, 15%).

### 3.2. Comparison of HPV Vaccine Intention, HPV Knowledge, Health Literacy, and Health Beliefs among Baseline and after the First and Second Intervention (Table 2)

At baseline, 70 participants (40%) in the LINE group and 76 participants (42%) in the Mail group reported that they had HPV vaccine intention sometime between “soon” and “within 3 years”, and no significant difference was observed. No significant difference in knowledge, literacy, and health belief model was consistently observed, both in PP and ITT. At the first intervention, there was no statistical difference in HPV vaccine intention, knowledge, literacy, and health belief model between the LINE and Mail groups. Although the data were not shown due to small numbers, the number of participants who answered that they got vaccinated was six (4%) and three (2%) in the LINE and Mail groups, respectively.

**Table 2 vaccines-10-02005-t002:** Comparison of HPV vaccine intention, knowledge, literacy, and health belief model at the first and second intervention.

	Baseline (*n* = 357)			First Intervention Assessment (*n* = 295)				Second Intervention Assessment (*n* = 278)			
	Total	LINE (*n* = 178, 50%)	Mail (*n* = 179, 50%)	LINE (*n* = 146, 49%)	Mail (*n* = 149, 51%)	*p **	OR (95% CI) *	LINE-Assisted Intervention(*n* = 142, 51%)	No Intervention (*n* = 136, 49%)	p *	OR (95% CI) *
	*n* (%)			No. of Participants/Total No. (%)				No. of Participants/Total No. (%)			
HPV vaccine intention	146 (41)	70 (40)	76 (42)	73/144 (51)	59/147 (40)			94/142 (66)	60/135 (44)		
Per protocol						0.079	1.52 (0.95–2.43)			>0.001	2.62 (1.59–4.35)
Intention to treat						0.108	1.47 (0.92–2.36)			0.002	2.08 (1.32–3.28)
Knowledge											
Median (Min–Max)	8 (6–10)	7 (6–10)	8 (6–11)	10 (8–13)	10 (8–12)			12 (10–14)	11 (9–13)		
≥ Median **	195 (55)	88 (49)	107 (60)	72/146 (49)	90/149 (60)			81/142 (57)	62/136 (46)		
Per protocol						0.059	0.64 (0.41–1.02)			0.059	1.58 (0.98–2.53)
Intention to treat						0.084	0.67 (0.42–1.06)			0.084	1.55 (0.94–2.55)
Literacy											
Median (Min–Max)	10 (6–14)	10 (6–14)	10 (6–15)	12 (8–16)	11 (8–15)			13 (10–16)	12 (10–16)		
≥ Median **	195 (55)	96 (54)	99 (55)	86/146 (59)	85/149 (57)			84/142 (59)	71/136 (52)		
Per protocol						0.740	1.08 (0.68–1.72)			0.253	1.32 (0.82–2.12)
Intention to treat						0.688	1.10 (0.70–1.72)			0.176	1.34 (0.88–2.05)
Health belief model											
Perceived susceptibility	151 (42)	68 (38)	83 (46)	79/145 (54)	81/147 (55)			90/142 (63)	79/135 (59)		
Per protocol						0.900	0.97 (0.61–1.54)			0.378	1.25 (0.76–2.04)
Intention to treat						0.832	1.05 (0.66–1.69)			0.291	1.29 (0.80–2.07)
Perceived severity	262 (74)	126 (71)	136 (76)	111/144 (77)	117/148 (79)			110/142 (77)	111/135 (82)		
Per protocol						0.695	0.90 (0.51–1.56)			0.329	0.75 (0.41–1.34)
Intention to treat						0.779	0.92 (0.53–1.60)			0.433	0.80 (0.45–1.41)
Perceived benefit	235 (66)	112 (64)	123 (69)	113/145 (78)	115/148 (78)			112/142 (79)	106/135(79)		
Per protocol						0.948	1.02 (0.59–1.77)			0.943	1.02 (0.58–1.81)
Intention to treat						0.703	1.12 (0.63–1.98)			0.839	1.06 (0.61–1.84)
Perceived barriers	49 (14)	25 (14)	24 (13)	24/143 (17)	18/147 (12)			19/139 (14)	20/134 (15)		
Per protocol						0.289	1.44 (0.73–2.83)			0.779	0.91 (0.45–1.82)
Intention to treat						0.261	1.48 (0.75–2.92)			0.995	1.00 (0.52–1.91)

* Logistic regression model stratified by sex. ** The items were divided binary by total median in each time. Multiple imputation was used to account for missing data.

At the second intervention, HPV vaccine intention in the LINE-assisted intervention group had a higher proportion of HPV vaccine intention than in the no intervention group: the odds ratios (OR, 95% confidence interval (CI)) were 2.62 (1.59–4.35) and 2.08 (1.32–3.28) in the PP and ITT analyses, respectively. Although the data were not shown due to small numbers, the number of participants who answered that they got vaccinated was four (3%) and five (4%) in the LINE-assisted intervention and no intervention groups, respectively. No statistical difference was observed in knowledge, literacy, and health belief model between the LINE-assisted intervention and no intervention groups.

We compared the percentage of people who answered all 20 knowledge questions correctly. At the baseline, “Cervical cancer can be prevented by using vaccines” in the LINE group had a significantly lower rate than in the Mail group (10% vs. 18%). At the first intervention, we found no significant difference between the LINE and Mail groups (Appendix A). At the second intervention, although there was no difference in the total score of the knowledge between the LINE-assisted intervention group and no intervention group, the intervention group had a significantly higher percentage than the no intervention group in items regarding “Cervical cancer screening is done by cytology”([OR: 1.78, 95% CI: 1.07–3.00] in PP and [OR: 1.89, 95% CI: 1.13–3.16] in ITT; Appendix A) and “Cervical cancer is caused by persistent infection with high-risk HPV” ([OR: 1.88, 96% CI: 1.51–3.07] in PP and [OR: 2.02, 95% CI: 1.28–3.19] in ITT; Appendix A).

When we compared the agreement rate of each item of the health belief models at the baseline, those of “I am worried about the side effects of the vaccine” in the LINE group was significantly higher than the Mail (60% vs. 50%) group, and those of “I do not know where I can get the vaccine” in the LINE group was lower than in the Mail group (51% vs. 62%). At the first intervention, those of “I am worried about the side effect of the vaccine” ([OR: 1.78, 95% CI: 1.10–2.86] in PP and [OR: 1.74, 95% CI: 1.08–2.83] in ITT; Appendix A) and “Three vaccinations is too much trouble” ([OR: 1.66, 95% CI: 1.03–2.67] in PP and [OR: 1.62, 95% CI: 1.01–2.59] in ITT; Appendix A) in the LINE group was significantly higher than in the Mail group, but these differences disappeared at the second intervention (Appendix A). At the second intervention, the agreement rate of “I do not know where I can get the vaccine” was significantly lower in the LINE-assisted intervention group than in the no intervention group ([OR: 0.61, 95% CI: 0.38–0.98] in PP and [OR: 0.67, 95% CI: 0.42–1.06] in ITT; Appendix A).

### 3.3. Number and Content of Dialogues in the LINE-Assisterd Intervention Group

In Round 1, there were 26 remarks: 11 remarks requested for information, 5 remarks requested for advanced information, 8 were thank-you notes for information, 2 remarks were a conversation between participants, and 1 sent information about HPV. In Round 2, there were 11 remarks: 6 remarks requested for information and 5 were thank-you notes for information. There were no disputes or moral hazards in either Round 1 or Round 2.

### 3.4. Comparison of HPV Vaccine Intention, HPV Knowledge, Health Literacy, and Health Belief Models between the Baseline and First Intervention

Although there was no difference in HPV vaccine intention between the LINE and Mail groups at the first intervention, the discordant rate was statistically different based on the McNemar test in the LINE group. Students in the LINE group were more likely to change their intention from “not intended” to “intended” one month after the first intervention (discordance rate, 29%, *mcnemar p* = 0.046; Appendix A). However, in the Mail group, the discordant rate was not significantly different (*mcnemar p* = 0.317; Appendix A).

Although there was also no difference in knowledge, literacy, and the health belief model between the LINE and Mail groups at the first intervention, the time series analysis demonstrated that the scores of HPV knowledge, health literacy, and perceived severity and benefit were significantly higher one month after the first intervention than at the baseline in both the LINE and Mail groups (*p* < 0.001, Appendix A; *p* < 0.02, Appendix A).

### 3.5. Comparison of HPV Vaccine Intention, HPV Knowledge, Health Literacy, and Health Belief Models between the First and Second Intervention

The discordant rate was statistically different based on the McNemar test in the LINE-assisted intervention group; these students were more likely to change their response from “not intended” at the first intervention to “intended” at the second intervention (49%, *mcnemar p* = 0.003; Appendix A). However, the discordant rate was not significantly different in the no intervention group (*mcnemar p* = 0.206; Appendix A).

In both the LINE-assisted intervention and no intervention groups, the time series analysis showed that the scores of health literacy at the second intervention were significantly higher than those at the first intervention (*p* < 0.01; Appendix A). In the LINE-assisted intervention group, the score of knowledge and susceptibility of health belief models was significantly higher than those at the second intervention (*p* < 0.009; Appendix A).

## 4. Discussion

This randomized study investigated whether LINE-assisted provision of information about HPV and cervical cancer prevention improves HPV vaccine intention, HPV knowledge, health literacy, and health belief models compared to the Mail (first intervention) or no intervention (second intervention) groups. At the first intervention, where PDF leaflets were provided, there were no significant differences in vaccine intention, knowledge, health literacy, and health belief models between the LINE and Mail groups. However, three months after the second intervention, which consisted of a seven-week provision of information, vaccine intention was significantly higher in the LINE-assisted intervention group than in the no intervention group. We have discussed our results based on the context of the COVID-19 pandemic in Japan and on previous studies.

We need to emphasize that our primary outcome was HPV vaccine intention and not the actual uptake. When this study was being conducted, COVID-19 vaccination was more aggressively recommended than HPV vaccination because the COVID-19 vaccine uptake was low, especially among the young generation, who were considered critical to ease the local pandemic spread. In Japan, we experienced the first national state of emergency, which was declared on 16 April 2020, and lasted until the end of May [35]. Subsequently, we started to recruit our participants in June 2020, but the governor of Akita prefecture asked all residents to self-quarantine and refrain from going beyond the prefecture boundary until 19 June 2020 [35]. Thus, it was not possible for our participants to look for the clinics to get the shot. Furthermore, they could not get the shot at their own expense because the government had suspended active recommendation during that time. Also at that time, in Japan, the quadrivalent HPV vaccine (4vHPV, Gardasil, MDS, Lyon, France) was available, but the price was JPY 45,000~60,000 for three doses. Because our previous study found that approximately 20% of the students in Akita University rated financial strain as their primary concern during the pandemic [35], it may be difficult to allocate financial resources for the HPV vaccine. Considering these limitations, HPV vaccine intention was still low, and only 6% of the participants reported immediate vaccine intention, while 61% reported no such intention or did not know [36]. Therefore, although direct comparison with previous studies using vaccination coverage as an outcome is difficult, such low intention among university and college students in Japan is likely to exist, even after the COVID-19 pandemic, unless a strong intervention is planned.

Among various social networking services, we used LINE as a communication platform to change vaccine intention and knowledge, literacy, and health belief model in this study. Previous studies [32,34,36,37,38], which used Facebook, Instagram, and Twitter as communication platforms, reported that the use of social networking service allows researchers to easily connect with students and their parents, who are key persons for HPV vaccination research. However, it should be noted that high accessibility does not always guarantee high HPV vaccine uptake. For example, a HPV vaccine campaign reached 155,110 adolescents and engaged 2106 adolescents, but only a few received the HPV vaccine [39]. Indeed, the usefulness of SNS to increase the level of HPV vaccine intention varies owing to limited evidence [39,40,41,42]. Among a few randomized controlled trials that reported a relatively higher level of scientific evidence, a Facebook-based RCT of college students reported an increase in awareness and knowledge of the HPV vaccine [43], while another Facebook-based RCT of parents found no difference in vaccination rates between the intervention and control groups [44]. Surprisingly, although this study was a descriptive panel report, which appears to have a lower level of scientific evidence, sending a leaflet from the local government of a small city in Japan improved vaccination rate [45]. In our study, we failed to show the superiority of LINE over Mail in relation to HPV vaccine intention. This lack of statistical significance may be explained by the fact that the median of knowledge was slightly higher in the Mail group than in the LINE group at baseline. Furthermore, the results of the time series analysis demonstrated that the scores for knowledge, literacy, and severity and benefit statistically increased in both the LINE and Mail groups. Thus, it was suggested that the tool for information dissemination does not matter in our study, and LINE does not increase the level of HPV vaccine intention level compared to Mail group.

For the second intervention, the LINE assisted seven-week provision of information significantly improved the HPV intention compared to the no intervention group. There was also no significant difference in knowledge, literacy, and health belief models between the intervention and no intervention groups. However, the time series analysis demonstrated that the scores of knowledge, literacy, and susceptibility significantly increased in the LINE-assisted intervention group, while only literacy scores significantly increased in the no intervention group. Such difference within each allocated groups may explain HPV vaccine intention. Furthermore, students in the LINE-assisted intervention group were allowed to freely make any comments and interact with other participants in the closed LINE chat room. Content analysis of the dialogues revealed that the majority of the interaction was dominated by one-way queries addressed to the administrative office, which was run by the Department of Environmental Health Science and Public Health, Akita University Graduate School of Medicine, and all participants knew that the members were medical students. Furthermore, Ireland and Denmark, whose HPV uptake previously plummeted to 50%, but successfully increased their immunization rate to 80% [45,46], attribute their recovery to the availability of a forum for direct contact with health care providers for girls and parents, through phone and online media [24,38,43,47]. Previous studies agree that the way in which healthcare professionals recommend vaccines has been associated with increased immunization coverage [6,13,48,49]. Medical assurance of vaccine safety and efficacy [50] and expert recommendation of vaccines [51] are factors that promote vaccine uptake, and healthcare professionals are thought to play an important role in the decision making regarding HPV vaccination among young adolescents [19]. Thus, it was suggested that the trust in health professionals may have played a role in our study.

There are some limitations that should be addressed. First, as our participants were limited to students belonging to the four universities in Akita prefecture, sampling bias may exist. In addition, because this study was conducted during the COVID-19 pandemic, when there was a state of emergency, the numbers of participants were relatively small. Second, participants who were more interested in HPV might have participated in our study, which may have led to the overestimation of HPV vaccine intention. Nevertheless, here, HPV vaccine intention was much lower than those previously reported in other Asian countries [52]. In fact, there were very few respondents who indicated immediate HPV vaccine intention. Third, the participants might have had higher IT literacy because we requested several works, including the electronic submission of informed consent with electronic signatures, creating an account in LINE application if a participant did not have an account, and joining the Open Chat in LINE. Fourth, although we used block randomization that was stratified by sex, factors such as faculty are still confounded. However, there was no significant difference in the proportion of faculty between the two groups [36]. Fifth, although the numbers were small, there were 12 students in the LINE group and 21 students in the Mail group who answered “Yes” to vaccine intention but changed to “No” after the first intervention. The same is true for the 13 students in the LINE-assisted group and 16 students in the no-intervention group who answered “Yes” during the first month after the first intervention survey and then answered “No” in the month immediately following the second intervention. We believe that this mind-change may have been caused by the negative information provided by our research team, pertaining to adverse effect, cost, and three doses. This type of behavior (i.e., mind-change after the disclosure of negative information) should be carefully handled in future studies. Sixth, although sexual factors regarding the number of partners, the gender of the partners, and sexual activities are relevant to vaccine intention, we did not include these questions, which consist of very private information, because we were very concerned that the inclusion of such private questions may have hampered participation rate. Seventh, we did not investigate how much our participants had an anti-vaccine view. In this regard, we asked about vaccination history about chickenpox, measles, mumps, and rubella. Of all the participants, there were only three who had never been exposed to these four vaccines (0.008%). This small number may indicate that there were very few participants in our study who had such anti-vaccine views. Thus, our results need to be interpreted carefully.

## 5. Conclusions

The present study demonstrated that there is no advantage in using LINE over Mail to provide HPV-related information to increase HPV vaccine intention. However, the LINE-assisted seven-week provision of information about the HPV vaccine and cervical cancer prevention may increase HPV vaccine intention, although the role of health professionals might influence HPV vaccine intention.

## Figures and Tables

**Figure 1 vaccines-10-02005-f001:**
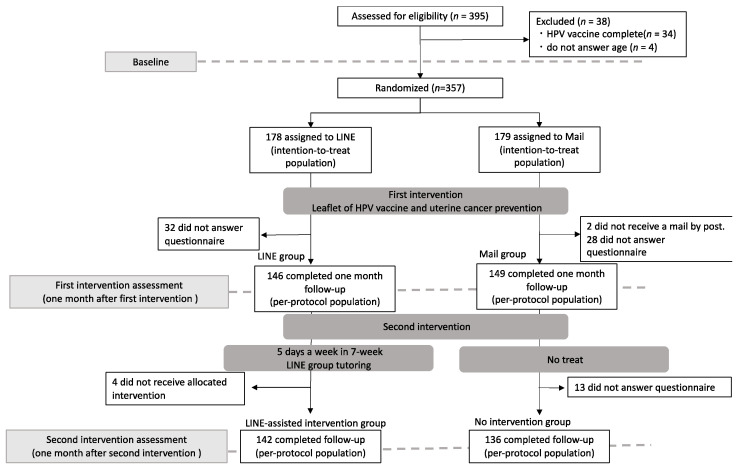
Flow diagram.

**Figure 2 vaccines-10-02005-f002:**
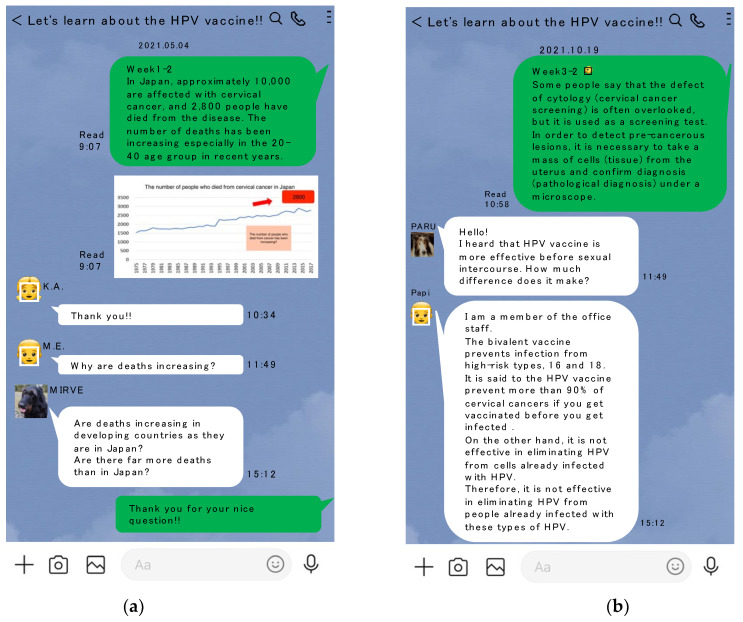
The image of the second intervention (reproduced for this article): (**a**) Round 1, (**b**) Round 2.

**Table 1 vaccines-10-02005-t001:** Baseline characteristics (*n* = 357).

		Total (*n* = 357)		HPV Vaccine Intention
		*n*	%	*p*
Sex	Female	189	53	0.334
	Male	168	47	
Age	<20	145	41	0.456
	≥20	212	59	
University	A	216	61	0.490
	B	65	18	
	C	65	18	
	D	11	3	
Faculty	Healthcare	147	41	0.406
	Not Healthcare	210	59	
Smoking	Never	339	95	0.439
	Now or ever	18	5	
Drinking	Rarely	242	68	0.491
	Frequently	115	32	
Sleeping	≥7 h	205	58	0.406
	<7 h	151	42	
Exercise (METS × hour/week)	≥Median (20 h)	179	50	0.829
	<Median (20 h)	178	50	
Breakfast	Everyday	157	44	0.524
	Not Everyday	200	56	
Reasons for not vaccinating (multiple choice)				
	Concerns of Side-Effect	86	24	-
	Parents Opposition	86	24	-
	I don’t know where to get vaccinated	79	22	-
	Expensive	53	15	-
	Sexually inactive	48	13	-
	Feel Troublesome	42	12	-
	I won’t get infected.	26	7	-
	Friends Not Vaccinated	24	7	-
	Only Checkup is enough	17	5	-
	Schools do not recommend	14	4	-
	Afraid of get infected by COVID-19	10	3	-
	Others	116	32	-

Numbers in each category that do not reach total numbers indicate missing data.

## Data Availability

The datasets generated and/or analyzed during the current study are available from the corresponding author on reasonable request.

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
