# Peer review of "Influence of LINE-Assisted Provision of Information about Human Papillomavirus and Cervical Cancer Prevention on HPV Vaccine Intention: A Randomized Controlled Trial"

_vaccines, 2022, doi:10.3390/vaccines10122005_

Round 1

Reviewer 1 Report

The authors present a study targeting HPV vaccination and investigated LINE and conventional email information to increase the vaccination rate among students. The authors concluded that LINE-based intervention increased vaccination intention and coverage. In the end, they discuss the multiple limitations of the study. The vaccine coverage is low also in other countries and social media-based approaches to increase knowledge and vaccination rates have been discussed multiple times. Here the authors tested such an approach and it turn out to be of limited success. However, conducting the study during lockdown may certainly have impacted the vaccination rates. Maybe it is worth running a second round addressing some of the limitations preferably not during lockdown.

The authors provided information on vaccination programs and recommendations. A comparison with other countries such as Australia would underline the importance of vaccination programs and the success and safety of the vaccines.

The introduction would benefit from at least a brief introduction about LINE. This would help non-LINE users to understand the intervention strategy.

The authors' intervention study targeted students while the initial recommendations target the younger individual of age 12-16, in other countries even younger. The authors could specify why they chose to invite students and not younger individuals and if the intervention study would facilitate vaccination in younger teenagers as well.

Line 116 on an extra point. Sentence structure.

The authors described in many details the intervention study and made efforts to maintain this blind non-biased one. Who were the chat partners of the participants? Students, supporting staff, and physicians, were there special training for the chat partners? The report would benefit from additional information about this.

Table 2 would benefit from lines to increase orientation. A horizontal display in landscape format would help to improve readability.

The authors could discuss why the "I am worried about the side effects of the vaccine" belief was higher in the LINE than mail group and if special training of the chat partner could prevent this.

The authors should specify the vaccination regime in Japan. HPV vaccination follows a 0-1-6 months scheme which means the three doses are necessary for full immunization. The authors should provide data or at least discuss if the higher vaccination intention in the LINE group is equivalent to a higher vaccination rate with a full immunization.

The authors should discuss previous intervention programs in Japan and other social media interventions regarding the HPV vaccination in other countries and compare results and success in more detail especially targeting the question if HPV vaccination is recommended for younger teenagers below 16 or even 12 in some countries.

The authors wrote in the abstract "The LINE-assisted provision of information was effective in improving HPV vaccine intention among unvaccinated Japanese university and college students" and in the discussion "at least LINE does not increase the level of HPV vaccine intention level compared to Mail group" and "at least LINE does not increase the level of HPV vaccine intention level compared to Mail group". In the current writing, the authors seem to contradict themselves in their statements.

Author Response

We deeply appreciate the editors and reviewers for their productive and helpful comments. Here we respond to each comment one by one in Word file.

Reviewer 2 Report

Thank you for inviting me to review this manuscript

In general, the aims are quite broad and will be challenging to exclude all other variables /influences to allow sufficient and robust conclusions

Not clear to an international audience exactly what LINE is, this should be clarified in one sentence at the beginning of the manuscript, same applies to MAIL.

Page 2

Line 49

Instead of “would like” consider “would choose”

Line 50

It has previously been “reported”

Line 61

Would avoid starting a sentence with And Line 70 Please be consistent is it Mail or mail?

Line 70

Should it be “increases” rather than “improves “

Would consider it better to use the term participant rather than entrees

Page 3 Line 116, seems to have a full stop in the middle of the sentence ?

Was there any follow up or description of those who did not complete the trial, relative to those who did?

Is there a copy of the questionnaire available (Google form-based survey) SEction 2.4.1 It is not clear to me why the answers of “immediately” all the way to “within three years” were collapsed into one group It seems unlikely that someone who has thought about doing something in the next three years is the same as someone who as given the response that they will do this immediately …how was this decision made and were there any references to support this?

2.4.2

Basic questions …according to whom? How was the basic versus more comprehensive information ascertained?

2.4.3

Are the HL questions validated?

2.4.4.

Are the questions used here previously validated, how were they chosen?

2.4.5

In some universities Science is separate to healthcare and I wonder was this considered in the co-variates as we know that healthcare students are likely to be more positive when it comes to health beliefs

Table 2 the * is not explained

Also terms like : feel troublesome and only checkup is enough should be made clearer

Page 10

Section 3.3.

Who monitored and gave the information and what were their qualifications? This should be included in methods section

Line 339; what were moral hazards defined as?

Line 401: SNS?
Line 414: would use the term lack of statistical significance rather than insignificant

Author Response

(The authors gave the same response as above.)

Reviewer 3 Report

Ota and colleagues have undertaken a survey to undersatnd how to best improve acceptance, and down the line vaccination, as it relates to HPV vaccination in Japan. HPV vaccination in Japan is a comlex area so this is definitely work of interest. However, the approach, methods and data presented are problematic with a number of issues needing resolution.

- There should be a clear statement of how HPV vaccination, approval and access vary between males and females. 47% of the cohort are male so this would have a major effect on the discussion.

- results need to be viewed in context of age. It may be good to increase vaccination in individuals over the age of 30 but this will have minimal effects on HPV infection and resultant cervical disease. data needs more detail in how the age of participants is relevant. these details can be descriptive rather than quantiative.

- cervical cancer is not uterine cancer

- the authors should describe why they think 12 - 13 individuals changed their intention to vaccinate from yes to no after interventions (S3/S4)

Figures S1, S2, and S4 have issues with the graphs. There is no y axis label and the bars of the same strength of signal and not equal, e.g. Figure S2 Barriers has four bars at 3.0 with two different heights

- the authors need to explain why they restricted the study to those of Japanese ethnicity. Being fluent in the language, and being a resident for a certain number of years are explainable but not ethnicity. Do the authors think that there is something unique, genetically, in Japanese ethnicity rather than culturally?

- could the authors identify whether participants were assessed for anti-vaccine views, as opposed to a lack of knowledge, prior to enrolement?

- Figure 2 has a number of errors, including that cervical screening, by cytology or any other methods, is for the identicaation and diagnosis of cevrical cancer. Cervical screening is to identify cervical lesiosns which are pre-cancerous so that they can be treated prior to becoming cancer. The fact that funadmental incorrect information is being given is probelmatic.

- There is a continued issue where it is unclear whether the information being presented is in relation to Japan only or whether it is international.

- Covariates are problematic and not explained. Smoking, drinking, breakfast and sleep are not explained to their relevance. Questions about sex, how many partners, the gender of the partners, sexual activities etc are much more relevant as it relates to HPV.

- it is not clear why the study did not fulfill the sample size described

- even if not statistically significant the p values should be included in the data rich tables.

- The 20 questions asked are deeply problematic. There does not appear to be a key with the correct answers in the manuscript see below for specific issues with these questions;

8 assuming this is specific to Japan as many countries now use HPV for screening

9 unclear on this question as unsure whether this is specific to Japan

13 - this questions is very problematic. HPV is an STI but the data suggests >80% of all individuals who have had sex have had a HPV infection at some point. HPV in those with another STI would be close to 100%.

17 - the ability of HPV vaccination to prevent cancer is dependent on the type of HPV ssociated cancer, when the vaccine was given, etc...

18 - the adverse events information swaps between whether it is flu or JEV. It is also worth noting that reported AEs are not representative of actual AEs

20 - cytology and Pap smear are the same thing

Author Response

We deeply appreciate the editors and reviewers for their productive and helpful comments. Here we respond to each comment one by one in the Word file.

Reviewer 4 Report

The authors conducted a prospective, randomized, parallel group, and open label trial to investigate whether the use of LINE would increase HPV vaccine intention among unvaccinated  university students. They concluded that there is no advantage in using LINE over mail to provide HPV-related information to increase HPV vaccine intention. 

the paper is suitable for publication after minor revision:

- modify the conclusion of abstract in according with the conclusion of paper.

- clarify  the role of person who respond to mail/line

Author Response

(The authors gave the same response as above.)

Round 2

Reviewer 2 Report

Much improved

Author Response

Thank you very much for providing important comments. We are thankful for the time and energy you expended.

Reviewer 3 Report

See highhlighted comments below in reference to the authors responses. In short, their study included 47% of people who couldn't get the HPV vaccine, did not stratify by age which is absolutely required to measure efficacy of their study, they didnt ask pertinent questions about sexual behaviour, and appeared to have given information in such a way as to scare a decent number of people away from vaccination

#1

So 47% of the participants in the study were not eligible for HPV vaccination at the time of the study? I am unsure the different between approval in December 2020 and the statement that men are not eligible, is this in reference to the (subsidized) screening program?

#2

This does not address the issue that there is no information on the age of the cohort other than the breakdown of <20 years of 20+ years. How people process information will vary depending on a range of factors including age. If the people how are becoming more likely to get vaccinated are >30 years of age then the benefits to disease outcome would be much less than if these people were< 20 years, or prior to any form of sexual contact.

#3

This is very concerning. If the study actually caused people to become less likely to vaccinate then it throws into question the study design and how information was presented. The presentation of ‘negative’ information indicates that it was not presented well. Adverse events (AEs) occur, however reported serious AEs are overwhelmingly found to either not be biologically and/or temporally plausibly caused by HPV vaccines, or even in a higher number of cases that you’d expect – to be real events.

#5

I can see the logic but would ask the authors to consider this for their next study- it is likely that the majority of ethnically Japanese people don’t understand why the program was halted. Language proficiency and/or years of residence (as in were they resident at the age of which vaccination was offered) could be reasonable inclusion criteria. I think that understanding how those with information from different countries view the Japanese recommendation could be a good control group as you would imagine that these individuals would be theoretically less likely to change their views – but knowing if your program does still have benefits for a more educated population could give an idea of its potential longevity

#7

Considering the unique situation for HPV vaccination in Japan this is a serious oversight

#8

The authors still miss the point. Cervical screening is not designed to detected cervical cancer.

Cervical screening is designed to detected pre-cancerous lesions (CIN1 – 3, AIS) which can be treated to reduce the incidence of cervical cancer

#10

These factors were not omitted for ethical reasons. The authors have stated that they were not asked because the authors view the cohort as being socially conservative. It is unclear whether the authors attempted to include these questions and were rejected by the HEC, or they made the presumption that people wouldn’t answer.

Author Response

Reviewer #3

Thank you very much for providing important comments. We are thankful for the time and energy you expended. Our responses to your comments are highlighted in blue. The newly revised text are now shown in red.

#1

So 47% of the participants in the study were not eligible for HPV vaccination at the time of the study? I am unsure the different between approval in December 2020 and the statement that men are not eligible, is this in reference to the (subsidized) screening program?

→No. Males were not eligible for the routine HPV vaccination during this study. Males could still receive the HPV vaccine before December 2020 but were able to get the shot at their own expense. Since December 2020,  Gardasil was approved for male vaccination but the shot requires males to have specific disease entity which meet the indication required for health insurance.

We have clarified that boys were not eligible for the routine HPV vaccination in Japan in Introduction.

Line 40 - 50

Under the Japanese Immunization Program in 2013, ……………. with a 0-2-6 months scheme [2]. Gardasil was approved for male vaccination in December 2020 [3]. However, as of now, men are not eligible for the routine HPV vaccination in Japan. Although the program for girls had been enacted, only a few months after the first reports of adverse reactions surfaced, the Ministry of Health, Labour, and Welfare decided to suspend active recommendation in 2013 [4].

#2

This does not address the issue that there is no information on the age of the cohort other than the breakdown of <20 years of 20+ years. How people process information will vary depending on a range of factors including age. If the people how are becoming more likely to get vaccinated are >30 years of age then the benefits to disease outcome would be much less than if these people were< 20 years, or prior to any form of sexual contact.

→ We added the 25th percentiles, median, and 75th percentiles of age. We also added the number of people under 26 and over 27 years of age, since the CDC states that HPV vaccination is recommended until the age of 26.

Line 310-312

The 25th percentiles, median, and 75th percentiles of age were 19 years, 20 years, and 22 years, respectively. The participant under the age of 26 were 349 (98%), and those over the age of 27 were 8.

#3

This is very concerning. If the study actually caused people to become less likely to vaccinate then it throws into question the study design and how information was presented. The presentation of ‘negative’ information indicates that it was not presented well. Adverse events (AEs) occur, however reported serious AEs are overwhelmingly found to either not be biologically and/or temporally plausibly caused by HPV vaccines, or even in a higher number of cases that you’d expect – to be real events.

→The reviewer’s question refers to previous #4 (NOT #3). We deeply agree with the reviewer’s concern that how information especially on negative information was presneted may affect their intention from “Yes” to “No” after the intervention. Considering that this is a sexually transmitted disease and it is hoped that more people will be vaccinated, we feel that these results need to be carefully examined. 

We have added the following sentences in the study limitation section.

Line 492-500

Fifth, although the numbers were small, there were 12 students in the LINE group and 21 students in the Mail group who answered "Yes" to vaccine intention but changed to "No" after the first intervention. The same is true for the 13 students in the LINE-assisted group and 16 students in the no-intervention group who answered "Yes" during the first month after the first intervention survey and then answered "No" in the month immediately following the second intervention. We believe that this mind-change may be caused by the negative information provided by our research team, pertaining to adverse effect, cost, and three doses. This type of behavior (i.e., mind-change after the disclosure of negative information) should be carefully handled in the future studies. 

#5

I can see the logic but would ask the authors to consider this for their next study- it is likely that the majority of ethnically Japanese people don’t understand why the program was halted. Language proficiency and/or years of residence (as in were they resident at the age of which vaccination was offered) could be reasonable inclusion criteria. I think that understanding how those with information from different countries view the Japanese recommendation could be a good control group as you would imagine that these individuals would be theoretically less likely to change their views – but knowing if your program does still have benefits for a more educated population could give an idea of its potential longevity

→The reviewer’s question refers to previous #6 (NOT #5). We deeply appreciate the reviewer’s comment on the inclusion of international students as a comparison study with Japanese ethnicity. We will definitely think of it in our next study.

#7

Considering the unique situation for HPV vaccination in Japan this is a serious oversight

→We assume that the comment from the reviewer pertained to anti-vaccine views. We actually asked about vaccination history about measles, rubella, mumps, chicken-pox. The participants who had never been exposed to these four vaccines were only 3 persons (0008%). This small number may indicate that there are very few participants who had such anti-vaccine view in our study.

Line 504 - 509

Seventh, we did not investigate how much our participants had an anti-vaccine view. In this regard, we asked about vaccination history about chickenpox measles, mumps, and rubella. The participants who had never been exposed to these four vaccines were only 3 persons (0008%). This small number may indicate that there are very few participants who had such anti-vaccine view in our study. Thus, our results need to be interpreted carefully.

#8

The authors still miss the point. Cervical screening is not designed to detected cervical cancer.Cervical screening is designed to detected pre-cancerous lesions (CIN1 – 3, AIS) which can be treated to reduce the incidence of cervical cancer

→Thank you for your advice. Cervical screening is not a test for cancer, and it's a test to help prevent cancer. So, we changed the Figure2.

#10

These factors were not omitted for ethical reasons. The authors have stated that they were not asked because the authors view the cohort as being socially conservative. It is unclear whether the authors attempted to include these questions and were rejected by the HEC, or they made the presumption that people wouldn’t answer.

→We were not rejected by the HEC but did not include the questions related to individual sexual behaviour. This was because we thought that some students did not disclose their sexual behaviour which is very private information because we were very concerned that the inclusion of such private questions may hamper participation rate.

In this round of revision, we have slightly modified the study limitation section as follows.

Line 500 - 504

Sixth, although sexual factors regarding the number of partners, the gender of the partners, and sexual activities are relevant to vaccine intention, we did not include these questions which is very private information because we were very concerned that the inclusion of such private questions may hamper participation rate.